# Surface-Enhanced Raman Spectroscopy Combined with Multivariate Analysis for Fingerprinting Clinically Similar Fibromyalgia and Long COVID Syndromes

**DOI:** 10.3390/biomedicines12071447

**Published:** 2024-06-28

**Authors:** Shreya Madhav Nuguri, Kevin V. Hackshaw, Silvia de Lamo Castellvi, Yalan Wu, Celeste Matos Gonzalez, Chelsea M. Goetzman, Zachary D. Schultz, Lianbo Yu, Rija Aziz, Michelle M. Osuna-Diaz, Katherine R. Sebastian, W. Michael Brode, Monica M. Giusti, Luis Rodriguez-Saona

**Affiliations:** 1Department of Food Science and Technology, The Ohio State University, Columbus, OH 43210, USA; nuguri.2@buckeyemail.osu.edu (S.M.N.); delamocastellvi.1@osu.edu (S.d.L.C.); wu.5671@buckeyemail.osu.edu (Y.W.); matosgonzalez.1@buckeyemail.osu.edu (C.M.G.); giusti.6@osu.edu (M.M.G.); rodriguez-saona.1@osu.edu (L.R.-S.); 2Department of Internal Medicine, Division of Rheumatology, Dell Medical School, The University of Texas, 1601 Trinity St., Austin, TX 78712, USA; 3Departament d’Enginyeria Química, Universitat Rovira i Virgili, Av. Països Catalans 26, 43007 Tarragona, Spain; 4Department of Chemistry and Biochemistry, The Ohio State University, Columbus, OH 43210, USA; chelsea.goetzman@srnl.doe.gov (C.M.G.); schultz.133@osu.edu (Z.D.S.); 5Savannah River National Laboratory, Jackson, SC 29831, USA; 6Center of Biostatistics and Bioinformatics, The Ohio State University, Columbus, OH 43210, USA; lianbo.yu@osumc.edu (L.Y.); william.brode@austin.utexas.edu (W.M.B.); 7Department of Internal Medicine, Dell Medical School, The University of Texas, 1601 Trinity St., Austin, TX 78712, USA; rija.aziz@austin.utexas.edu (R.A.); michelle.osuna@austin.utexas.edu (M.M.O.-D.); kate.sebastian@austin.utexas.edu (K.R.S.)

**Keywords:** long COVID, fibromyalgia, surface-enhanced Raman spectroscopy, volumetric absorptive micro-sampling, dried bloodspot cards

## Abstract

Fibromyalgia (FM) is a chronic central sensitivity syndrome characterized by augmented pain processing at diffuse body sites and presents as a multimorbid clinical condition. Long COVID (LC) is a heterogenous clinical syndrome that affects 10–20% of individuals following COVID-19 infection. FM and LC share similarities with regard to the pain and other clinical symptoms experienced, thereby posing a challenge for accurate diagnosis. This research explores the feasibility of using surface-enhanced Raman spectroscopy (SERS) combined with soft independent modelling of class analogies (SIMCAs) to develop classification models differentiating LC and FM. Venous blood samples were collected using two supports, dried bloodspot cards (DBS, *n* = 48 FM and *n* = 46 LC) and volumetric absorptive micro-sampling tips (VAMS, *n* = 39 FM and *n* = 39 LC). A semi-permeable membrane (10 kDa) was used to extract low molecular fraction (LMF) from the blood samples, and Raman spectra were acquired using SERS with gold nanoparticles (AuNPs). Soft independent modelling of class analogy (SIMCA) models developed with spectral data of blood samples collected in VAMS tips showed superior performance with a validation performance of 100% accuracy, sensitivity, and specificity, achieving an excellent classification accuracy of 0.86 area under the curve (AUC). Amide groups, aromatic and acidic amino acids were responsible for the discrimination patterns among FM and LC syndromes, emphasizing the findings from our previous studies. Overall, our results demonstrate the ability of AuNP SERS to identify unique metabolites that can be potentially used as spectral biomarkers to differentiate FM and LC.

## 1. Introduction

Long COVID (LC) is a heterogenous clinical syndrome that persists or develops beyond the acute phase of COVID-19 infection, termed as the post-acute sequelae of SARS-CoV-2 infection (PASC) [1,2]. COVID-19 typically lasts for 2–3 weeks; however, around 10–20% of patients remain symptomatic after initial recovery [3,4,5], persisting for more than 4 weeks, according to the Center for Disease Control and Prevention [6,7]. Furthermore, LC has also been found to emerge in a small cohort following the administration of SARS-CoV-2 vaccines [8]. Although epidemiologic data indicate that many LC patients experience significant symptom resolution within a year, a substantial portion of patients do not, and these individuals experience LC as a chronic, having relapsing and remitting symptoms over time [8,9,10]. Commonly reported symptoms include sleep disturbances, autonomic dysregulation, widespread pain, fatigue, post-exertional malaise, and physical and cognitive dysfunction [8,11,12,13]. Possible mechanisms include the progressive effect of acutely damaged tissues, chronic inflammation leading to the onset of immune dysregulation, autoimmunity, and endothelial dysfunction [14]. Although research is uncovering common pathophysiological mechanisms associated with LC, the absence of a consensus definition and the heterogeneity of clinical presentations complicate its diagnosis [6,14]. Fibromyalgia (FM) is a chronic central sensitivity syndrome characterized by augmented pain processing at generalized body sites, affecting nearly 5% of the world’s population, predominantly women [15,16,17]. FM presents as a multimorbid clinical condition, often accompanied by sleep disturbances, mood disorders, chronic fatigue, intestinal irritability, migraines, and temporomandibular joint disorder [18]. FM is characterized by the amplification of neuronal signals leading to central sensitization, resulting from an imbalance in the central nervous system and neuroendocrine function [19]. FM patients experience chronic non-malignant pain, which shares similarities with the pain experienced by individuals with LC. Due to the overlapping symptoms and poorly defined underlying pathogenesis of these syndromes, accurately diagnosing and distinguishing between FM and LC poses challenges [6,20]. Moreover, patients are often treated with unnecessary opioids to alleviate the pain, leading to suboptimal treatment outcomes and resource drain [6,16]. These clinically similar syndromes negatively impact individual’s participation and quality of life, underscoring the need for a diagnostic method based on metabolic biomarkers.

Vibrational spectroscopy (IR and Raman) captures unique chemical signatures of global metabolites within biological samples, including potential biomarkers. This technology enables disease diagnosis in a relatively short time with minimal sample pre-processing and is advancing across various medical domains such as rheumatology, cancer, and urology [21,22,23]. Surface-enhanced Raman spectroscopy (SERS) has been widely used in biomedical applications due to its high detection sensitivity and significant signal enhancement, ranging from 10^5^ to 10^14^ times the Raman intensities [24,25]. SERS can detect trace biomarkers in samples with detection limits ranging approximately from ng/mL to fg/mL [26,27]. Moreover, its low susceptibility to water interferences and minimal sample requirements makes it ideal for analyzing biological fluids [6]. The spectral data produced by SERS yield large amounts of information, which are analyzed using multivariate pattern recognition techniques. Soft independent modeling of class analogy (SIMCA) is a supervised classification methodology where the syndrome status corresponding to each spectrum is provided, along with the spectral data. SIMCA constructs a classification model by carefully assessing the differences in spectral information among different groups, potentially originating from associated biomarkers [18].

Our research group has been extensively assessing the vibrational spectroscopic characteristics of FM and related rheumatologic conditions (rheumatoid arthritis (RA), chronic low back pain (CLBP), systemic lupus erythematosus (SLE), and osteoarthritis (OA)) over the past decade [6,18,21,28,29,30]. We have identified the potential of portable mid-infrared (MIR), MIR micro-spectroscopy, FT-Raman micro-spectroscopy, and SERS in diagnosing FM from similar symptomatic syndromes (RA, SLE, OA, CLBP). Peptide backbones, aromatic acids, and carboxylic acids have been shown to be important in discriminating between these groups.

In a comprehensive study using MIR, we focused on improving the extraction methodology of low molecular fractions (LMFs) from the blood samples and observed that washed filter-based protocol outperformed unwashed semi-permeable filter membranes and direct measurements of blood aliquots [30]. Subsequently, in the following study [6], we extended our knowledge of sample handling and MIR measurements for metabolomic analyses of clinically similar conditions, LC and FM, obtaining perfect classification with 100% accuracy. Interestingly, we identified a distinct spectral biomarker at 1565 cm^−1^ in FM patients compared to LC subjects through spectral deconvolution of pre-processed spectra. This signature was potentially associated with the side chain of glutamate amino acids. In our next investigation for FM diagnosis using SERS [29], we performed a detailed analysis to optimize the synthesis of gold nanoparticles (AuNPs) compatible with metabolites and developed a standard operating procedure (SOP) for measuring the LMFs [29]. In these studies, venous blood samples were collected on dried bloodspot (DBS) cards due to their storage feasibility and reduced blood volume [31]. However, in DBS cards, blood with varying hematocrit levels diffuses at different rates, leading to sample volume inhomogeneity. To address this, Neoteryx Mitra^®^ device was selected, which collects blood using a novel approach that absorbs a fixed volume of blood onto a porous substrate, known as volumetric absorptive micro-sampling (VAMS) [32].

The objective of this research was to explore the potential of using SERS spectroscopic technique combined with pattern recognition analysis to develop a classification algorithm to discriminate FM and LC patients. Blood samples were collected with DBS cards and VAMS tips to evaluate their effects on the Raman spectral signals.

## 2. Materials and Methods

### 2.1. Patient Sample Recruitment and Sample Storage

All studies involving human subjects were approved by The University of Texas at Austin institutional review board and abided by the Declaration of Helsinki principles. Following IRB approval (study no. 2020030008)/approval date 19 June 2020) and informed consent, blood samples were obtained from patients with LC and FM at the University of Texas at Austin Post Covid Program and Fibromyalgia and Central Sensitivity Syndrome clinics located at University Texas Health Austin Clinics, Austin, Texas. Bloodspots on LC subjects were obtained between November 2022 and March 2024. Bloodspots on patients with FM and healthy controls (NC) were obtained from September 2020 through March 2025. Patients’ blood samples were collected and stored either on bloodspot cards (Whatman 903Blood Protein Saver Snap Apart Card, GE Healthcare, Westborough, MA, USA) at −20 °C (*n* = 48 FM, *n* = 46 LC and *n* = 4 NC) or samples were collected intravenously and stored on Neoteryx Mitra devices (Neoteryx, CA, USA) with a 30 µL total collection volume tip employing VAMS technology (*n* = 39 FM, *n* = 39 LC and *n* = 11 NC). Each device was equipped with two tubes sampled from the same patient, providing analytical replicates. All samples were then shipped on dry ice to the Rodriguez-Saona’s Vibrational Spectroscopy laboratory at The Ohio State University and kept at −20 °C until extraction for subsequent analysis. Standardized circles on the filter paper serve as a guide to ensure collection of the bloodspot size was standardized by collecting samples on cards with preprinted circles as guides, with each circle containing approximately 50 µL of blood per spot.

Questionnaires: All subjects with FM provided a self-report of symptoms through the use of the validated questionnaires. All subjects with FM were given the Revised Fibromyalgia Impact Questionnaire (FIQR), a 10-item self-rating instrument that measures physical functioning, work status, depression, anxiety, sleep, pain, stiffness, fatigue, and wellbeing [33]. The Beck Depression Inventory (BDI) is a 21-item, self-report rating inventory that measures characteristic attitudes and symptoms of depression [34]. The Symptom Impact Questionnaire Revised (SIQR) is the FM-neutral version of the FIQR and does not assume the patient has FM [35]. The SIQR was utilized as a measure of physical functioning, work status, depression, anxiety, sleep, pain, stiffness, fatigue, and wellbeing of all subjects with LC. The Central Sensitization Inventory (CSI) is a two-part patient-reported outcome measure that assesses somatic and emotional symptoms common to CSS [36]. The McGill Pain questionnaire (MPQ) is an instrument providing descriptive aspects of pain as well as pain intensity [37].

Criteria for the diagnosis of FM included the following: age 18–80 with a history of FM and meeting current criteria for FM [38]. To become a patient at the Post-COVID-19 Program, individuals were required to have reached a minimum of 12 weeks from the onset of their initial COVID-19 illness. Documentation of a positive COVID-19 test was not required to receive services, although all patients were screened by a dedicated nurse to validate that their clinical history was consistent with a prior COVID-19 infection, and the symptoms are likely associated with PASC and not an obvious alternative etiology. Sigmaplot v15.0 and SigmaStat v4.0 software (Inpixon, Palo Alto, CA, USA) were utilized for statistical analysis of questionnaires.

### 2.2. Sample Extraction

For blood sample extraction, a procedure used in previous studies [6,18,29,30] based on washed-filter-membrane was incorporated. Amicon^®^ Ultra centrifugal filters (Merck Millipore, Billericia, MA, USA) with 10 kDa cut-off were used to fractionate low-molecular-weight (LM) solutes from the blood samples. Before extraction, the filters were rinsed with 3 mL Milli-Q water, followed by centrifugation at 4000 rpm for 10 min (Figure 1). This step ensured the removal of glycerol present on filter membranes that could produce artifacts in the SERS spectra and interfere with multivariate analysis [18].

Two blood collection systems, dried bloodspot cards (DBS, Whatman 903 Blood Protein Saver Snap Apart Card, GE Healthcare, Westborough, MA, USA) and volumetric absorptive micro-sampling tips (VAMS, Neoteryx, CA, USA) were used. The DBS card or the VAMS tip was placed in 1 mL Milli-Q water under sterile conditions and sonicated (15 min for VAMS and 30 min for DBS) to efficiently extract the blood samples from the specimen collector. The blood solutions were transferred into the washed filters and centrifuged at 4000 rpm for 15 min at 4 °C. Small molecules permeated through the filters while high-molecular fractions were retained on the filters. Overall, the centrifugal force and semi-permeable membrane enriched the LM fractions (LMFs) containing water-soluble solutes important for biomarker identification [39]. The permeated solutes were dried using nitrogen-flushing and vacuum drying (4 h at room temperature) to obtain the LMF film (Figure 1). The extracted samples were stored at −80 °C and were analyzed by SERS within 2 days of extraction to prevent any signal loss due to sample degradation. Additionally, process blank samples were prepared using blank (unused) DBS cards and VAMS tips to evaluate the contribution of sampling substrate and processing factors, including filters, and the VAMS tips were also used to evaluate the SERS signal.

### 2.3. AuNP Synthesis and Characterization

Gold nanoparticles (AuNPs) were synthesized using the citrate reduction method [29,40]. Prior to use, glassware and stir bars were immersed in a sulfuric acid solution of Alnochromix^TM^ (Alconox laboratories, White Plains, NY, USA) overnight and then rinsed with boiling ultrapure water. Briefly, 0.061 g of HAuCl_4_.H_2_O was added to 500 mL Milli-Q water in a conical flask with constant stirring (350 rpm) at 90 °C for 30 min. The sodium citrate solution, prepared by dissolving 0.059 g of sodium citrate in 7.5 mL water, was carefully added to the solution under stirring conditions. The resulting AuNP dispersion changed to red color in a few minutes, indicating the formation of monodisperse spherical particles [29], and was left stirring until cooled to room temperature. The AuNPs were stored in glass vials (Restek, Center county, PA, USA) at room temperature, protected from light until further use.

AuNPs were characterized using ultraviolet–visible (UV-Vis) extinction spectroscopy and dynamic light scattering (DLS). An Agilent 8453 UV-Vis spectrometer (Agilent Technologies, Santa Clara, CA, USA) was used to obtain extinction spectra from AuNPs, resulting in an average absorption maximum at 540 nm (Appendix A). DLS and zeta potential measurements were performed using a Malvern Panalytical Zetasizer (Malvern Panalytical Ltd., Westborough, MA, USA) equipped with a 532 nm laser. The average particle size of the synthesized AuNPs was 33 nm (see Appendix A) with an average zeta potential of −14 ± 17.7 mV.

### 2.4. Sample Preparation for SERS Analysis

For SERS measurements, 1 mL of synthesized AuNPs were centrifuged for 25 min at 6000 rpm, and the supernatant was decanted from the pellet to remove excess citrate [41]. The pelleted particles were resuspended in 1 mL of a 1:1 mixture of acetonitrile and water. Ten microliters of this suspension was thoroughly mixed with the dried sample, and 5 µL was transferred onto the aluminum-covered well slide (BRAND company Inc., South Hamilton, MA, USA). Transmission electron microscope (TEM) images of AuNPs suspended in water and acetonitrile (1:1) and AuNPs mixed with LFM extract from FM subject were taken using a FEI-Technai G2-30 electron microscope.

### 2.5. SERS Instrumentation

SERS Raman spectra were acquired using an Optical PhotoThermal InfraRed (OPTIR) Raman micro-spectrometer (Photothermal Corporation, Santa Barbara, CA, USA). The 785 nm optical laser probe provided the Raman excitation with an output power of 97 mW. The laser was transmitted through a line filter and focused to a spot size of 0.6 µm on the sample, with a 50× objective lens having a numerical aperture of 0.8. Measurements were collected from 600 cm^−1^ to 2300 cm^−1^ with a spectral resolution of 2 cm^−1^. The calibration of the Raman system was verified using a 520 cm^−1^ vibrational band of silicon standard. PTIR studio software (version 4.5.1, Photothermal Corporation, Santa Barbara, CA, USA) was used to control spectral acquisition. Our previous study demonstrated the conditions optimal for acquiring SERS signal [29]. Collecting one SERS Raman spectrum with 20 s exposure time at 8.7 mW of the laser power was found to give higher intensity. Multiple spectra (2 to 6) were obtained from different locations on the sample well. The spectra for individual sample were carefully assessed and filtered for chemometric analysis based on their spectral profile and intensities.

### 2.6. Pattern Recognition Analysis

After SERS measurement, individual spectra were baseline-corrected using the Rubber band baseline correction algorithm offered by PTIR studio software (Version 4.5.1, Photothermal Spectroscopy Corp., Santa Barbara, CA, USA). Pattern recognition analysis was performed on the baseline-corrected spectral data in a chemometrics modeling platform, Pirouette version 4.5 (Infometrix Inc., Woodville, WA, USA).

A supervised classification algorithm, soft independent modeling by class analogy (SIMCA), was employed to distinguish FM spectra from LC. SIMCA develops principal component (PC) models for each training category. It assigns class of the unknown sample by projecting the data into PC space of each class and determining the class that best fits [42,43]. Eighty percent of the dataset was used as training set and the remaining 20% was used as test set. The dataset was mean-centered, and pre-processing techniques, including normalization, smoothing (7-point window), and second derivative, were applied during the multi-variate analysis. Mean centering facilitates a comparison of inter-variable associations in lieu of absolute values [42]. Normalization standardizes the response intensities to a value of 100. Smoothing reduces high-frequency noise and improves the chemical signals, while the second derivative deconvolutes overlapping peaks, recognizing information relevant to class separation, and reducing baseline errors [18]. Important spectral regions were selectively considered, containing information relevant to describing the class separation by removing the noisy signals resulting in enhanced model performance. The optimal number of factors were determined when total residual X-variance (SECV) was minimized. A leverage vs. studentized residual plot was used to identify outliers. Pirouette software determined the critical boundary in the plot based on Mahalanobis distance. Samples beyond these critical values were carefully assessed, and those with a high difference from the remaining data points were excluded [42].

The leave-one-out cross-validation approach (LOOCV) was used to internally validate the calibration model. In this method, each sample from the training set is excluded and estimated by the model developed using the remaining calibration samples. The process is iterated until all the samples have been left out and predicted. Performance metrics including sensitivity, specificity, and accuracy were computed using the misclassification table of the calibration and validation set, which summarized the classification success by groups [42]. Furthermore, the prediction results of the test set were used to generate a receiver operating characteristic (ROC) plot using the R computational platform [44], incorporating the pROC package (v1.18.5) [45]. The area under the curve (AUC) of the ROC plot was computed using the DeLong method [46,47], with a 95% confidence interval. The AUC assessed the diagnostic accuracy of the model.

## 3. Results

### 3.1. Clinical Characteristics of Subjects

The clinical characteristics of the patients with FM and LC are presented in Table 1 and Table 2. Table 1 shows the characteristics of FM and LC subjects who were analyzed via blood cards. FM subjects with (*n* = 46, F:46, M:0) had a mean age of 41.9 ± 13.0. Their BMI was 32.1 ± 8.7, with a mean FIQR of 55.6 ± 20.3. The BDI was 20.7 ± 11.5, the CSI was 61.5 ± 15.3, and the MPQ was 99.4 ± 51.3. Patients with LC (*n* = 46, F:27, M:19) had a mean age of 48.4 ± 14.5. Their BMI was 29.3 ± 8.0 with a mean SIQR of 44.4 ± 22.3. NC subjects with (*n* = 4, F:2, M:2) had a mean age of 32.5 ± 14.8. Their BMI was 27.1 ± 6.0, with a mean SIQR of 6.7 ± 7.1. The BDI was 5.8 ± 8.0, the CSI was 25.8 ± 16.6, and the MPQ was 9.0 ± 10.2. Table 2 shows the characteristics of FM and LC subjects who were analyzed via VAMS technic. FM subjects with (*n* = 38, F:38, M:0) had a mean age of 44.1 ± 14.0. Their BMI was 31.3 ± 7.9, with a mean FIQR of 49.7 ± 19.5. The BDI was 18.3 ± 11.7, the CSI was 61.9 ± 16.9, and the MPQ was 91.3 ± 46.1. Patients with LC (*n* = 38, F:22, M:16) had a mean age of 51.1 ± 13.8. Their BMI was 29.5 ± 7.9 with a mean SIQR of 46.4 ± 22.3. NC subjects with (*n* = 11, F:9, M:2) had a mean age of 48.5 ± 18.7. Their BMI was 23.1 ± 4.0, with a mean SIQR of 2.7 ± 3.4. The BDI was 3.1 ± 3.8, the CSI was 17.1 ± 11.7, and the MPQ was 3.3 ± 4.2.

### 3.2. Interaction of AuNPs with LMF Extract

TEM image of the AuNP suspension (Figure 2a) demonstrates that the AuNPs were spherical in shape. The presence of nanoparticles and their interaction with LMF extract was further confirmed by the transmission electron microscopy image (Figure 2b). Further, the TEM image of AuNPs with the sample (Figure 2b) displayed the presence of heterogenous LM fractions with AuNPs aggregated to different extract fragments. SERS presents the molecular fingerprinting capabilities of Raman spectroscopy with increased sensitivity due to the effective amplification of the inherent Raman modes of molecules in close proximity to the plasmonic nanostructure [48]. The plasmon resonance condition and surrounding medium, including the morphology of nanoparticles, define the spectral position of the plasmon mode [48]. The intensity of SERS signals is influenced by different enhancement mechanisms (primarily, electromagnetic enhancement, and chemical enhancement) occurring in the sample. This depends on various factors, including the nanoparticles, interaction between the molecules and the nanoparticles, orientation of Raman active modes towards the nanostructured surface, and the field components, among others [48].

### 3.3. Process Blank

Figure 3 compares the dried bloodspot card (DBS) and VAMS absorbent tips process blanks. The DBS process blank (Figure 3a) exhibited contributions at 1356 cm^−1^, 1397 cm^−1^, and 1562 cm^−1^, likely associated with C-H bending, υ(COOH) and υ(CN), and C=C bending, respectively, from the wax and pectic constituents of the Whatman card support [49]. These results agree with other studies that demonstrated the signal contribution from the filter paper can affect the measurements of metabolites and elements, particularly present at low concentrations, as assessed by quadrupole time-of-flight mass spectrometry [50] and inductively coupled plasma mass spectrometry [51]. On the contrary, the VAMS process blank (Figure 3b) displayed no prominent SERS bands and a very low response signal, indicating minimal to no contribution from the substrate and sample processing conditions.

### 3.4. Spectral Analysis

The spectra were screened based on their spectral profile and signal intensity. The measurements ranged from 600 to 2300 cm^−1^, although the region of 1700 2300 cm^−1^ contained no prominent signals and was excluded prior to the data analysis.

Figure 4 shows the averaged normalized spectra (Figure 4a) and its corresponding pre-processed spectra (smoothed and second-derivatized, Figure 4b).

The SERS bands from VAMS samples were well defined and had a prominent signal in the amide I region (1600–1670 cm^−1^) compared to spectra from DBS, as highlighted in Figure 4a. When comparing the transformed spectra from the two blood collection supports, a distinct profile was observed in the region around 1245 cm^−1^ to 1320 cm^−1^ (Figure 4b, highlighted in blue), while additional bands were observed in the VAMS spectra (Figure 4b, highlighted in green) at 1332 cm^−1^ and 1637 cm^−1^ (associated to amide I). Furthermore, the spectral pattern of the VAMS samples remained consistent regardless of the signal intensity (Appendix A). The Raman shifts corresponding to significant bands identified in Figure 4b were linked to several biochemical components and molecular structures (Table 3) based on the reported literature [24]. Altogether, the signals captured by the SERS explained contributions from aromatic amino acids, peptides, and lipids.

The averaged spectra of normalized intensities were used to compare the spectral differences among the LC, FM, and NC groups obtained from VAMS tips and DBS cards. Compared with the mean spectra of NC (Figure 5a), the FM spectra of VAMS tips showed increased bands at 1092, 1160, 1278, 1549, and 1617 cm^−1^, and decreased bands at 1016, 1227, and 1447 cm^−1^. The LC group (Figure 5b) showed positive deviation at bands 1090, 1156, 1278, 1566, and 1608 cm^−1^ and negative deviation in areas 1017, 1227, and 1461 cm^−1^. Overall, the subtracted spectra of NC signals from FM and LC of VAMS tips demonstrated similar and comparable profiles.

Subtle variations were observed in the subtracted mean spectrum between the FM and LC groups of VAMS samples (Appendix A). The subtracted spectrum resulted an increase in bands at 1019, 1136, 1363, and 1464 cm^−1^, and a decrease at 1322 and 1559 cm^−1^ in the LC class when compared to the FM class.

Compared with mean spectral intensities of NC samples, the FM (Figure 6a) and LC (Figure 6b) of DBS showed an increase at bands 1119 and ~1215 cm^−1^, and a decrease at band 1363 cm^−1^. Further, the subtracted mean spectra of LC from FM (Appendix A) exhibited fluctuations at 1160, 1227, 1447, 1532, 1634, and 1667 cm^−1^. Overall, the differences in the band intensities of LC, FM, and NC groups can be assigned to the molecular structures of proteins (amide I, amide III, aromatic, and acidic amino acids), CH_2_ stretching, and carboxylate stretching.

### 3.5. Pattern Recognition Analysis

#### 3.5.1. Classification Models Based on VAMS Collection Support

The differences in the spectral data between LC and FM disorders were evaluated using a supervised classification analysis, SIMCA. SIMCA conducts PCA for samples belonging to each class. The class is assigned based on the residual distance, determined by projecting new observations into the PC space of individual class [66]. Three regions of the SERS spectra were considered to create the classification models: full spectral range (700–1700 cm^−1^), central region (1100–1730 cm^−1^), and a region focused on the amide bands (1400–1700 cm^−1^). The data were normalized and the replicates of each class were averaged. Two classes SIMCA models were developed using the averaged dataset. Less than 15% spectra had borderline characteristics with other disorders and were eliminated. Subsequently, 80% of the averaged dataset was used to construct the calibration model and the remaining 20% was used as unseen data to externally validate the generated model.

Seven to eight latent variables (LVs) were used that cumulatively captured 90–92% of the variation in the dataset. Selecting an appropriate number of LVs is essential to achieve a model that balances between underfitting and overfitting, ensuring accurate predictions [18]. A lower number of factors may not describe sufficient variance, causing underfitting, while an excessive number of descriptors could introduce random noise, leading to overfitting of the model [67,68]. Additionally, various classification objects, including discriminating power plot, Cooman’s plot, and performance metrics, rely on these latent variables. Table 4 provides details on sample size, applied pre-treatments, selected LVs, cumulative variation explained by the LVs, and variables important in discriminating the groups in the calibration models.

Figure 7 illustrates the discriminating power plot and Cooman’s plot corresponding to the calibration model incorporating the central region (1100–1730 cm^−1^). A higher value in the discriminating power plot implies a greater discrimination ability in the variable [42]. In Figure 7a, 1353 cm^−1^ demonstrates the highest discriminating power. The variables important in discriminating the disease groups are listed in Table 4. Overall, the wavenumbers 858 cm^−1^ (ring bending), 1220–1280 cm^−1^ (amide III, CH stretching), 1353 cm^−1^ (COO^−^ stretching, CH deformations and C=C stretching) [69], 1412 cm^−1^ (indole rings), 1439 cm^−1^ (CH_2_ groups) [70], and 1563 cm^−1^ (aromatic groups, COO^−^ stretching) suggest potential contributions from peptides, aromatic, and acidic amino acids.

Cooman’s plot visually represents the classification information by plotting the class distance against each other [42,71]. The classification model clustered the samples into two groups (Figure 7b) resulting in 96% accuracy, 100% sensitivity, and 93% specificity (Table 4). Sensitivity evaluates the capability of the model to accurately classify LC cases in a population with LC disease, while specificity assesses the proportion of correctly identified FM subjects within a population with FM disease [18]. Accuracy estimates the ability of the model to correctly classify the subjects into their respective groups [18]. The discriminating power plot and Cooman’s plot for other SIMCA models have been provided in Appendix A. SIMCA uses the Cooman’s plot to evaluate the classification performance by assigning the class in terms of the distance from the model [71]. The two threshold lines define critical distances corresponding to a 95% probability threshold.

Table 5 summarizes the classification performance of the calibration and validation models including the three spectral regions. Sensitivity evaluates the capability of the model to accurately classify LC cases in a population with LC disease, while specificity assesses the proportion of correctly identified FM subjects within a population with FM disease [18]. The models exhibited similar performance in the calibration set, with performance measures ranging from 89% to 100%. External validation with unseen data is essential to verify the statistical significance of the separation [18]. When validated with an independent test set, the model incorporating the central spectral region (1100–1730 cm^−1^) provided the best validation results with 100% accuracy, sensitivity, and specificity, presumably due to less random signals originating from 700 to 1000 cm^−1^, and the presence of important information contributed by various functional groups. The model including the full spectral region (700–1700 cm^−1^) demonstrated a good performance with 86% accuracy, sensitivity, and specificity. However, the protein region (1400–1700 cm^−1^) provided 100% sensitivity but lower accuracy (71%) and sensitivity (43%) compared to the other models, possibly due to the exclusion of important functional groups modeling the characteristics of LC disease.

The ROC visualizes the sensitivity and specificity of the model’s performance over the range of possible decision thresholds [72]. It summarizes the overall diagnostic accuracy of the model by providing area under the curve (AUC) values. The AUC values range from 0 to 1, where a value of 1 reflects a perfectly accurate prediction and 0 implies inaccurate test [72]. The calibration model gave an AUC greater than 0.90, exhibiting an “outstanding” classification accuracy [72]. The validation set corresponding to the central region (1100–1730 cm^−1^) achieved an excellent AUC of 0.86 with 0.63 to 1 95% confidence interval (Figure 8), followed by 700–1700 cm^−1^ (AUC = 0.73) and 1400–1700 cm^−1^ (AUC = 0.67).

#### 3.5.2. Classification Models Based on DBS Collection Support

SIMCA models were built and analyzed similarly for the data acquired from DBS cards. The region around 1353 cm^−1^, which exhibited a noticeable contribution from the DBS card, was excluded to avoid variations introduced by the support matrix in the data analysis. Two regions of the SERS spectra of DBS samples were considered to create the classification models: full spectral range (700–1330 cm^−1^ and 1400–1700 cm^−1^) and central region (1100–1330 cm^−1^ and 1400–1700 cm^−1^). Samples having borderline characteristics were identified and removed from the dataset. A total of 80% of the averaged dataset was used to train the model and 20% was used as the test set for external validation. Seven to eight latent variables (LVs) were used that collectively described 78–90% of the variation in the dataset. Table 6 presents information on sample size, applied pre-treatments, selected LVs, cumulative variation explained by the LVs, and important variables for discriminating between groups in the calibration models developed using the DBS training set.

Figure 9 illustrates the discriminating power plot and Cooman’s plot corresponding to the DBS calibration model incorporating full spectral ranges (700–1330 cm^−1^ and 1400–1700 cm^−1^). Variables 735, 1287, and 1639 cm^−1^ demonstrated higher discriminating ability in the full spectral range model (Figure 9a). The classification model clustered the samples into two groups in the Cooman’s plot (Figure 9b), resulting in 100% accuracy, sensitivity, and specificity (Table 7). The discriminating power plot and Cooman’s plot for SIMCA models have been provided in Appendix A.

Table 7 summarizes the performance metrics of the calibration and validation models incorporating the full spectral region and the central region. The models exhibited similar performance in the calibration set, with performance measures ranging from 94% to 100%. Validating with an independent dataset revealed the best validation results by the model incorporating the full spectral region, resulting in 93% accuracy, 83% specificity, and 100% sensitivity. The model entailing the central region (1100–1330 cm^−1^ and 1400–1700 cm^−1^) provided acceptable results with 67% accuracy, 89% specificity, and 45% sensitivity. The calibration model showed an AUC greater than 0.85, exhibiting an “excellent” classification accuracy [72]. The validation set corresponding to the full spectral range (700–1330 cm^−1^ and 1400–1700 cm^−1^) achieved an acceptable AUC of 0.79 [72] with 0.54 to 1 95% confidence interval.

## 4. Discussion

Our team has extensively explored the capabilities of vibrational spectroscopy in diagnosing FM [6,21,28,30]. In our initial study, we demonstrated the ability of portable FT-MIR and Raman micro-spectroscopy to discriminate FM (*n* = 50) from other rheumatic disorders (RA = 29, OA = 19, SLE = 23) and reported that protein backbones and pyridine-carboxylic acids dominated the discrimination [28]. A follow-up study was conducted using SERS with AuNPs to discriminate FM from non-FM disorders (RA, SLE, OA, and CLBP) and found that spectral regions related to amino acids were associated with the discrimination [29]. We further extended our investigation analyzing samples collected using a novel VAMS technique to minimize hematocrit-related sample-volume inconsistencies [18], obtaining an excellent classification of FM vs. non-FM with 84%, 83%, and 85% accuracy, sensitivity, and specificity, respectively.

In this study, the discrimination between LC and FM subjects was explained by amide bands and amino acids allowing for the metabolomic analysis of clinically similar conditions with perfect classification accuracy [6]. Furthermore, through spectral deconvolution processing, we identified a distinct spectral band at 1565 cm^−1^ only present in FM subjects linked to the side chains of glutamate.

We further evaluated the ability of SERS with AuNPs in combination with pattern recognition analysis to investigate the potential biomarkers in classifying FM and LC samples collected using two different supports, DBS cards and VAMS tips. One of the major challenges associated with DBS is heterogeneity in sample volume due to different hematocrit (red blood cells per unit volume of blood) levels, subsequently impeding the quantitative analysis [73,74]. VAMS wicks a fixed volume of blood independent of hematocrit value, avoiding its impact on quantification [75,76]. Several studies analyzing VAMS have illustrated the high correlation of small molecules to the venous blood compared to DBS [77,78,79,80,81]. The DBS process blank revealed noticeable contributions from the wax and pectic constituents of the Whatman card support, while the VAMS tip process blank produced no significant response. Thus, the variability observed in the VAMS samples stems from the LMF extract and lacks the random variations presented by the collection matrix. Trifonova et al. (2019) observed variations in metabolomic profiles among different DBS collection materials, which they attributed to interfering contaminants from the paper cards, when evaluated using quadrupole time-of-flight mass spectrometry [50]. Additionally, Pederson et al. (2017) indicated that the signal contribution from the filter paper can impact the measurements of elements present at low concentrations [51].

SERS spectra of VAMS samples exhibited a prominent signal in the amide I region (1600–1670 cm^−1^). The transformed spectra of VAMS samples revealed additional bands at 1332 cm^−1^ and 1637 cm^−1^, providing the 2-class-SIMCA algorithm with additional data for disease classification. The subtracted spectra of NC signals from FM and LC of VAMS samples demonstrated similar and comparable profiles, although subtle variations linked to the molecular structures of peptides and amino acids were observed in the subtracted mean spectra of the LC from the FM group. In the VAMS 2-class-SIMCA model, the training data incorporating the central spectral region (1100–1730 cm^−1^) gave the best validation results with 100% accuracy, sensitivity, and specificity and an excellent classification accuracy with an AUC of 0.86. This could presumably be due to the elimination of random signals produced from 700 to 1000 cm^−1^, and the presence of important information contributed by various functional groups present in the central region (1100–1730 cm^−1^). Amide bands and aromatic and acidic amino acids were important in describing the separation of disease groups, emphasizing the findings of our previous studies. Conversely, the DBS calibration model incorporating the full spectral range (700–1330 cm^−1^ and 1400–1700 cm^−1^) achieved validation results of 93% accuracy, 83% specificity, 100% sensitivity, and an AUC of 0.79. This could possibly be attributed to the exclusion of the essential-information-containing region around 1353 cm^−1^, due to random noise introduced by the DBS substrate. Overall, VAMS samples offer a reliable model due to the absence of contributions from the collection matrix, consistent signal patterns, higher intensity, important information from the amide I, and better classification performance compared to DBS cards.

The clinically similar syndromes, FM and LC, have overlapping symptoms and undefined underlying pathogenesis, posing a challenge for accurate diagnosis. The cohort is often misdiagnosed despite following the revised criteria for FM due to significant errors stemming from the high subjectivity levels in the questionnaire [21]. As no reproducible biomarker is known for FM syndrome, affected individuals and physicians are eagerly looking for an objective evaluation and standardized references for diagnosis [21,82]. Our studies have been aiming towards developing a reliable diagnostic tool and determining potential spectral biomarkers for advancing treatment strategies. Overall, our findings underscore the ability of vibrational spectroscopy to identify unique metabolites important in diagnosing FM from other related diseases.

## 5. Conclusions

SIMCA models built using VAMS samples outperformed the models developed using DBS cards with a validation performance of 100% accuracy, sensitivity, and specificity, achieving an excellent classification accuracy of 0.86 AUC. VAMS substrate did not generate Raman signals, providing a reliable model compared with DBS cards. Analysis using IR and Raman, along with advances in the methodologies and instrumentation, facilitates our understanding on the underlying biochemical differences among the diseases, advancing FM diagnosis and treatment approach.

## Figures and Tables

**Figure 1 biomedicines-12-01447-f001:**
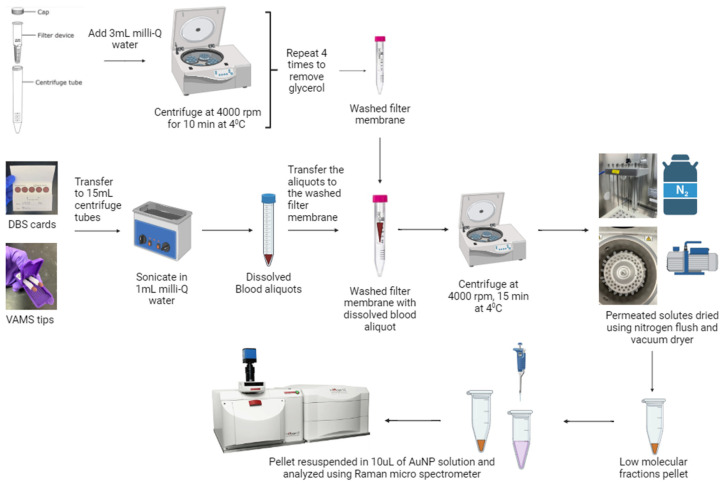
Illustration of extraction protocol of LMFs from blood aliquots collected on blood cards and VAMS tips.

**Figure 2 biomedicines-12-01447-f002:**
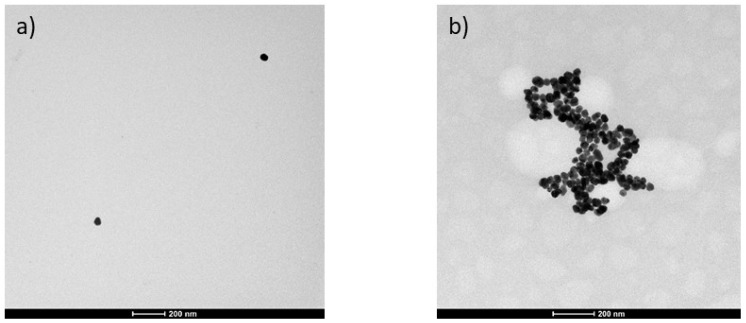
TEM images of AuNPs suspended in water and acetonitrile (1:1) (**a**) and AuNPs suspended in water and acetonitrile (1:1) mixed with the fractionate low-molecular-weight (LM) extract from FM sample stored in VAMS tip (**b**).

**Figure 3 biomedicines-12-01447-f003:**
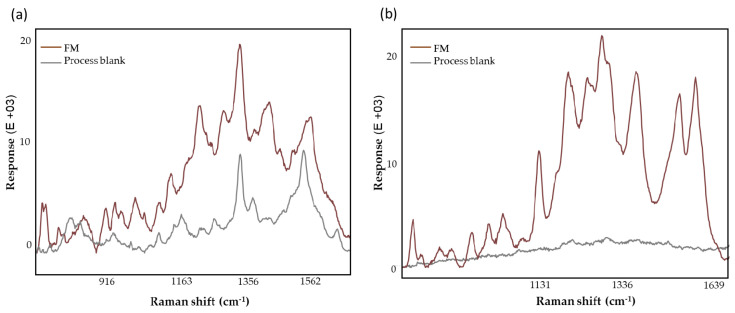
SERS spectra of (**a**) DBS and (**b**) VAMS process blanks (gray spectra) and the LMF (FM, brown spectra).

**Figure 4 biomedicines-12-01447-f004:**
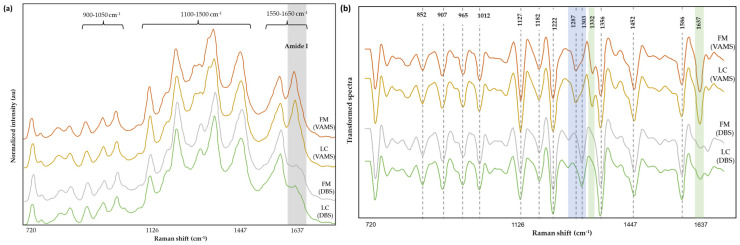
Normalized averaged spectra of FM and LC subjects (**a**) and transformed spectra (smoothing and second derivatization) (**b**) of samples collected using VAMS and DBS supports. The blue highlighted region indicates the different profile observed between VAMS and DBS supports, while the green highlighted region denotes the additional bands detected in the VAMS samples.

**Figure 5 biomedicines-12-01447-f005:**
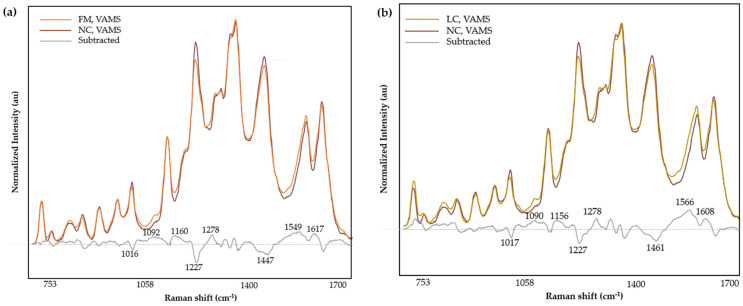
Normalized and averaged SERS spectra of LMF blood samples from FM (*n* = 36) and NC subjects (*n* = 11) (**a**), and LC (*n* = 38) and NC (*n* = 11) (**b**) collected on VAMS tips and the difference spectrum calculated from the mean SERS spectra among FM and NC subjects.

**Figure 6 biomedicines-12-01447-f006:**
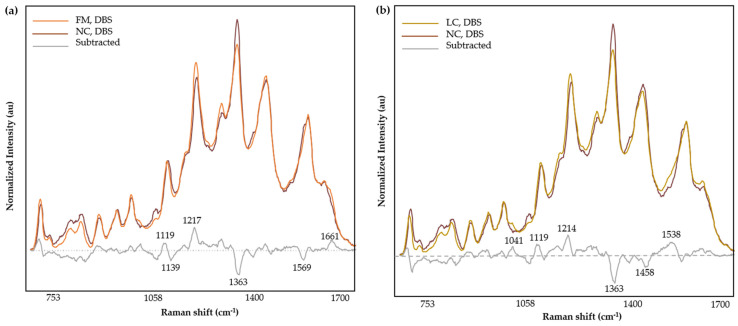
Normalized and averaged SERS spectra of LMF blood samples from FM (*n* = 33) and NC subjects (*n* = 4) (**a**) and LC (*n* = 42) and NC (*n* = 4) (**b**) collected in DBS cards and the difference spectrum calculated from the mean SERS spectra among FM and NC subjects.

**Figure 7 biomedicines-12-01447-f007:**
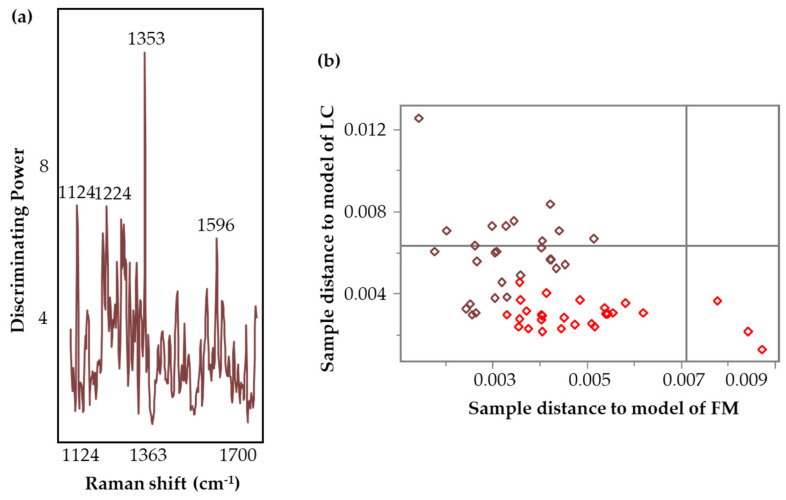
Discrimination power (**a**) and Cooman’s plot (**b**) (brown squares denote FM and red squares represent LC samples) of the calibration 2-class SIMCA model incorporating the central region (1100–1730 cm^−1^) for the classification of FM and LC samples evaluated by SERS Raman spectroscopy.

**Figure 8 biomedicines-12-01447-f008:**
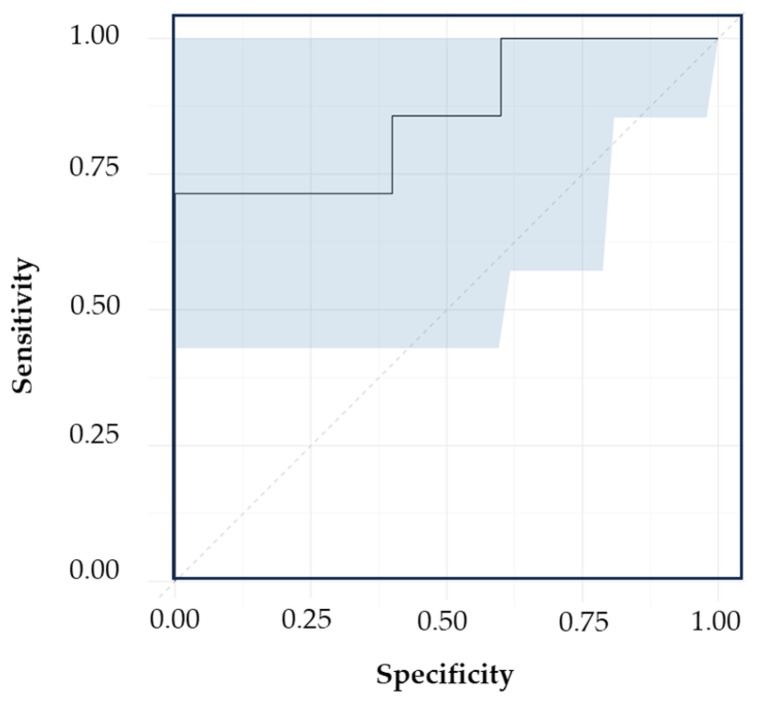
Validation ROC of the model incorporating the central region (1100–1730 cm^−1^), showing 95% confidence interval (shaded in blue) of the AUC (solid line).

**Figure 9 biomedicines-12-01447-f009:**
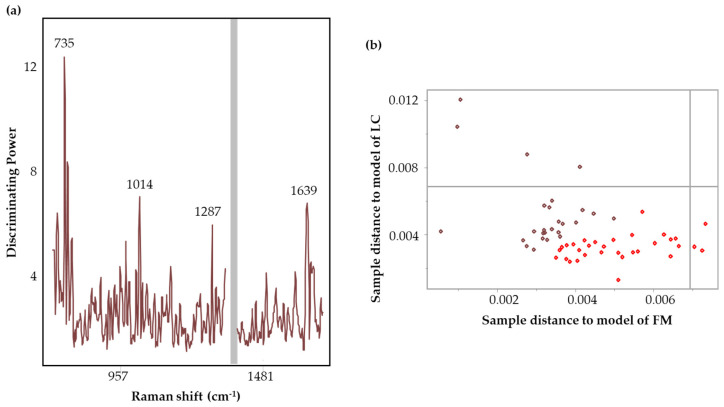
Discrimination power (**a**) and Cooman’s plot (**b**) (brown squares denote FM and red squares represent LC samples) of the calibration SIMCA model incorporating full spectral range (700–1330 cm^−1^ and 1400–1700 cm^−1^) for the classification of FM and LC samples evaluated by SERS spectroscopy.

**Table 1 biomedicines-12-01447-t001:** Clinical characteristics of all blood card subjects. Values expressed as mean ± sd; *n*: number of subjects, age (range). LC: Long COVID; FM: fibromyalgia, BMI: body mass index. FIQR: Revised Fibromyalgia Impact Questionnaire. SIQR: Symptom Impact Questionnaire Revised. BDI: Beck Depression Index. CSI: Central Sensitization Inventory, MPQ: McGill Pain Questionnaire, NC: healthy control.

	Age	*n* (M/F) [%M/%F]	BMI	FIQR	SIQR	BDI	CSI	MPQ
FM	41.9 ± 13.0	46 (0/46) [0/100]	32.1 ± 8.7	55.6 ± 20.3		20.7 ± 11.5	61.5 ± 15.3	99.4 ± 51.3
LC	48.4 ± 14.5	46 (19/27) [41/59]	29.3 ± 8.0		44.4 ± 22.3			
NC	32.5 ± 14.8	4 (2/2) [50/50]	27.1 ± 6.0		6.7 ± 7.1	5.8 ± 8.0	25.8 ± 16.6	9.0 ± 10.2

**Table 2 biomedicines-12-01447-t002:** Clinical characteristics of all volumetric absorptive micro-sampling (VAMS) subjects. Values expressed as mean ± sd; *n*: number of subjects, age (range). LC: Long COVID; FM: fibromyalgia, BMI: body mass index. FIQR: Revised Fibromyalgia Impact Questionnaire. SIQR: Symptom Impact Questionnaire Revised. BDI: Beck Depression Index. CSI: Central Sensitization Inventory, MPQ: McGill Pain Questionnaire, NC: healthy control.

	Age	*n* (M/F) [%M/%F]	BMI	FIQR	SIQR	BDI	CSI	MPQ
FM	44.1 ± 14.0	38 (0/38) [0/100]	31.3 ± 7.9	49.7 ± 19.5		18.3 ± 11.7	61.9 ± 16.9	91.3 ± 46.1
LC	51.1 ± 13.8	38 (16/22) [42/58]	29.5 ± 7.9		46.4 ± 22.3			
NC	48.5 ± 18.7	11(2/9) [18/82]	23.1 ± 4.0		2.7 ± 3.4	3.1 ± 3.8	17.1 ± 11.7	3.3 ± 4.2

**Table 3 biomedicines-12-01447-t003:** Band assignments of prominent SERS vibrational bands.

Band (cm^−1^)	Mode	Contributions	Reference
852	out-of-plane ring bending	Tyr	[52,53]
907	υ(COC)		[54]
965	ring breathing mode	Aromatic amino acids	[55]
1012	aromatic ring breathing vibrations, υ(C-N)	Phe	[56,57]
1127	υ(C-N), υ(C-C), NH_3_ deformation	Proteins, lipids	[58,59]
1182	δ(N-H)	Protein	[60]
1222	υ(C-H), amide III	Amino acids, proteins	[61]
1287	α-helix (amide III), CH_2_ wag	Trp, proteins	[59]
1303			
1332	υ(C-H)	Nucleic acids, phospholipids	[56,58,62]
1356	out-of-plane bending vibrations of H-C-H	Trp, nucleic acids	[56,63]
1452	δ(CH_2_)	Proteins, lipids, fatty acids	[58]
1586	ν(C=C), ν(COO), amide II	Aromatic and acidic amino acids, nucleic acids, proteins	[58,59,64,65]
1637	C=O stretching, α-helix (Amide I)	Proteins	[54,59]

**Table 4 biomedicines-12-01447-t004:** Information (sample distribution, transformations, LV, cumulative variation, and discriminating variables) related to the calibration models, incorporating the three spectral regions.

Spectral Region (cm^−1^)	Sample Distribution		Transformations	LV	Cumulative Variation	Discriminating Variables (cm^−1^)
	Cal.	Val.				
700–1700	28 FM 30 LC	8 FM 8 LC	SM (7), SD (19), MC	7	90%	858, 1224, 1248, 1281
1100–1730	27 FM 28 LC	7 FM 8 LC	SM (7), SD (13), MC	7	90%	1353
1400–1700	26 FM 28 LC	7 FM 7 LC	SM (7), SD (17), MC	6	92%	1412, 1439, 1563

SM, SD, MC, LV, Cal, and Val are smoothing, second derivative, mean centering, latent variable, calibration, and validation, respectively.

**Table 5 biomedicines-12-01447-t005:** Calibration and validation performance (accuracy, sensitivity, and specificity) of the SIMCA model classifying FM and LC specimens, collected using VAMS support.

Spectral Region (cm^−1^)	Calibration				Validation			
	AC	SP	SN	AUC	AC	SP	SN	AUC
700–1700	95%	89%	100%	0.94	86%	86%	86%	0.73
1100–1730	98%	100%	96%	0.96	100%	100%	100%	0.86
1400–1700	98%	96%	100%	0.96	71%	100%	43%	0.67

AC, SP, SN, are AUC are accuracy, specificity, sensitivity, and area under the curve, respectively.

**Table 6 biomedicines-12-01447-t006:** Information (sample distribution, transformations, LV, cumulative variation, and discriminating variables) related to the calibration models, incorporating two spectral regions, used in the development of DBS models.

Spectral Region	Sample Distribution		Transformations	LV	Cumulative Variation	Discriminating Variables (cm^−1^)
	Cal	Val				
Full region	26 FM 33 LC	7 FM 9 LC	SM (7), SD (13), MC	7	83%	735, 1287, 1639
Central region	38 FM 33 LC	10 FM 9 LC	SM (7), SD (13), MC	8	90%	1117, 1385, 1483, 1666

**Table 7 biomedicines-12-01447-t007:** Calibration and validation performance (accuracy, sensitivity, and specificity) based on two spectral regions’ full spectral range (700–1330 cm^−1^ and 1400–1700 cm^−1^) and central region (1100–1330 cm^−1^ and 1400–1700 cm^−1^) of the SIMCA model classifying FM and LC collected using DBS cards.

Spectral Region	Calibration				Validation			
	AC	SP	SN	AUC	AC	SP	SN	AUC
Full region	100%	100%	100%	0.88	93%	83%	100%	0.79
Central region	94%	95%	94%	0.87	67%	89%	45%	0.53

AC, SP, SN, are AUC are accuracy, specificity, sensitivity, and area under the curve, respectively.

## Data Availability

The data presented in this study are available on request from the corresponding author. The data are not publicly available due to privacy concerns.

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
