# Peer review of "Surface-Enhanced Raman Spectroscopy Combined with Multivariate Analysis for Fingerprinting Clinically Similar Fibromyalgia and Long COVID Syndromes"

_biomedicines, 2024, doi:10.3390/biomedicines12071447_

Round 1

Reviewer 1 Report

Comments and Suggestions for Authors

This study presents a serious, in-depth approach for discriminating between patients with fibromyalgia (FMG) and long COVID by blood sample analyses of small molecule profiles, which is a novel approach that presents such discrimination and should be appreciated. One point that disturbs this reviewer is, however the basis for making this comparison on the first place. Why should such molecular changes in life-long FMG patients be compared to long-COVID patients who are expected to recover, even if later than others? An expansion of the logical reasoning for such a comparison, and the link to the current approach will be helpful to the more biomedically-oriented readers, who are interested in the comparison but may have difficulties to understand its reasoning.

Author Response

Biomedicines Reviewer responses

Thank you for your review of our manuscript. We particularly would like to thank the reviewers for their thoughtful critiques. We have amended the manuscript accordingly and have highlighted those changes below.

Reviewer 1

  • One point that disturbs this reviewer is, however the basis for making this comparison on the first place. Why should such molecular changes in life-long FMG patients be compared to long-COVID patients who are expected to recover, even if later than others? An expansion of the logical reasoning for such a comparison, and the link to the current approach will be helpful to the more biomedically-oriented readers, who are interested in the comparison but may have difficulties to understand its reasoning.

FM and LC are functionally limiting conditions that involve chronic pain and fatigue, suggesting common pathways of central sensitization and neuroimmune dysregulation. FM is well-recognized as a chronic condition; however, as a new illness the natural history of LC is still being researched. Although epidemiologic data indicates that many LC patients experience significant symptom resolution within a year, a substantial portion of patients do not, and these individuals experience LC as a chronic similar to Myalgic Encephalomyelitis/Chronic Fatigue Syndrome (ME/CFS) and may having relapsing and remitting symptoms over time. (Ballouz et al., 2023; Davis et al., 2023; Hartung et al., 2024). Following the reviewer comment, we have added the following information in the Introduction section (lines 54 to 57) “Although epidemiologic data indicates that many LC patients experience significant symptom resolution within a year, a substantial portion of patients do not, and these individuals experience LC as a chronic condition having relapsing and remitting symptoms over time.”

Reviewer 2

  • The authors should add the TEM image of the Au NPs.

We agree with the reviewer. We have included in the manuscript a TEM image of our AuNPs mixed with water and acetonitrile (please see Figure 2 a). We have also added a section in the materials and methods explained how this image was obtained (lines 226 to 229) “Transmission electron microscope (TEM) images of AuNPs suspended in water and acetonitrile (1:1) and AuNPs mixed with LFM extract from FM subject, were taken using a FEI-Technai G2-30 electron microscope.”

  • If the Au NPs aggregate after mix the targets with the Au NPs solution. TEM image and UV-Vis spectra should be added.

We have also included in the manuscript a TEM image of our AuNPs aggregates after mixing LMF extract from a FM subject with our AuNPs suspended in water and acetonitrile (please see Figure 2 b). We have commented the TEM images in the results section as well (lines 316 to 320)” TEM image of the AuNP suspension (Figure 2 a) demonstrate that the AuNPs were spherical in shape. The presence of nanoparticles and their interaction with LMF extract was further confirmed by transmission electron microscopy image (Figure 2 b). Further, the TEM image of AuNPs with the sample (Figure 2b) displayed the presence of heterogenous LM fractions with AuNPs aggregated to different extract fragments.”. Unfortunately, the final volume that we have when we mix our LMF extract with our suspension of AuNPs is 10 µL. With our plate reader we need at least 100 µL to obtain a reliable reading and with 1:10 dilution the absorbance that we could obtain for the UV-Vis measurement is too low.

  • The authors should explain clearly of the SERS spectra. Such as the reason of the frequency of the SERS bands, and some SERS bands changed wider, and the intensity variation.

 We have added more information in the section 3.2 Interaction of AuNPs with LMF extract (lines 320 to 329) to better explain our SERS spectra differences “SERS presents the molecular fingerprinting capabilities of Raman spectroscopy with increased sensitivity due to effective amplification of the inherent Raman modes of molecules in close proximity to the plasmonic nanostructure [49]. The plasmon resonance condition and surrounding medium, including the morphology of nanoparticles, define the spectral position of the plasmon mode [49]. The intensity of SERS signals is influenced by different enhancement mechanisms (primarily, electromagnetic enhancement and chemical enhancement) occurring in the sample. This depends on various factors, including the nanoparticles, interaction between the molecules and the nanoparticles, orientation of Raman active modes towards the nanostructured surface, and the field components, among others [49].”

Reviewer 2 Report

Comments and Suggestions for Authors

The authors explored the SERS spectroscopic technique combined with pattern recognition analysis to develop a classification algorithm to discriminate FM and LC patients. Blood samples were collected with DBS cards and VAMS tips to evaluate their effects on the Raman spectral signals. This is very interesting research, I recommend it is publishable after the following revision.

Other comments:

1.      The authors should add the TEM image of the Au NPs.

2.      If the Au NPs aggregate after mix the targets with the Au NPs solution. TEM image and UV-Vis spectra should be added.

3.      The authors should explain clearly of the SERS spectra. Such as the reason of the frequency of the SERS bands, and some SERS bands changed wider, and the intensity variation.   

Author Response

(The authors gave the same response as above.)

Round 2

Reviewer 1 Report

Comments and Suggestions for Authors

The manuscript is now suitable for publication.